# The Value of Non-Invasive Optimal Vessel Analysis Quantitative Magnetic Resonance Angiography for Studying Flow and Collateral Patterns in Patients with Bilateral Carotid Steno-Occlusive Disease

**DOI:** 10.3390/brainsci15020211

**Published:** 2025-02-19

**Authors:** Fiona Helg, Elisa Colombo, Corinne Inauen, Lara Maria Höbner, Martina Sebök, Tilman Schubert, Jorn Fierstra, Antonio Spinello, Susanne Wegener, Andreas R. Luft, Zsolt Kulcsar, Luca Regli, Giuseppe Esposito

**Affiliations:** 1Department of Neurosurgery, Clinical Neuroscience Center, University Hospital Zurich, 8091 Zurich, Switzerland; fiona.helg@uzh.ch (F.H.); elisa.colombo@usz.ch (E.C.); lara.hoebner@usz.ch (L.M.H.); martina.seboek@usz.ch (M.S.); jorn.fierstra@usz.ch (J.F.); antonio.spinello@usz.ch (A.S.); luca.regli@usz.ch (L.R.); 2Faculty of Medicine, University of Zurich, 8006 Zurich, Switzerland; susanne.wegener@usz.ch (S.W.); andreas.luft@usz.ch (A.R.L.); 3Department of Neurology, Clinical Neuroscience Center, University Hospital Zurich, 8091 Zurich, Switzerland; corinne.inauen@usz.ch; 4Department of Neuroradiology, Clinical Neuroscience Center, University Hospital Zurich, 8091 Zurich, Switzerland; tilman.schubert@usz.ch (T.S.); zsolt.kulcsar@usz.ch (Z.K.); 5Cereneo Center for Neurology and Rehabilitation, 6354 Vitznau, Switzerland

**Keywords:** stroke, steno-occlusive disease, NOVA-qMRA

## Abstract

**Background/Objectives**: Bilateral steno-occlusive disease of the internal carotid artery (ICA) carries an increased stroke risk with associated high morbidity and mortality. Management of these patients is often complex. In this study, we evaluate the value of non-invasive optimal vessel analysis quantitative magnetic resonance angiography (NOVA-qMRA) for studying flow and collateral patterns in patients with bilateral carotid steno-occlusive disease. **Methods**: Patients with bilateral ICA-stenosis ≥ 50% who received NOVA-qMRA were included in this study. The volume flow rates (VFRs) of the A2-segment of the anterior cerebral artery (A2-ACA), M1-segment of the middle cerebral artery (M1-MCA), and P2-segment of the posterior cerebral artery (P2-PCA) were analyzed. Demographic, clinical, and treatment data were collected. **Results**: Twenty-two patients (mean age ± SD: 68 ± 10 years) were included. Nineteen patients (86%) were symptomatic. Thirteen patients (59%) were revascularized; among them, M1-VFR was significantly lower (*p*-value = 0.01) on the side selected for revascularization (88 mL/min ± 53) compared to the contralateral one (130 mL/min ± 56). P2-VFR was significantly higher (*p*-value = 0.04) in the treated subgroup (108 mL/min ± 41) than in the non-treated one (83 mL/min ± 34). **Conclusions**: The present study supports the use of NOVA-qMRA to study flow and collateral patterns in patients with bilateral steno-occlusive carotid disease, especially M1- and P2-VFR. This information may be helpful for decision-making and to tailor revascularization treatment.

## 1. Introduction

Stroke represents a major burden to the healthcare system with it being the second leading cause of death globally and a relevant cause of morbidity [1]. Bilateral ICA-stenosis and/or -occlusion is bearing a poor prognosis with an increased risk of stroke and consecutive disability, coma, and death [2].

Therapeutic management for this condition varies and is debated [3]. Given the variable symptomatology of bilateral steno-occlusive carotid disease, clinical presentation must be supported by further parameters to tailor the therapeutic strategy [4,5,6].

Magnetic resonance angiography (MRA) represents a well-established tool to study stroke patients and large cerebral artery stenosis or occlusion by using time-of-flight (TOF) sequences [7,8,9,10]. Unlike traditional methods that often rely on invasive procedures, contrast agents, or ionizing radiation, non-invasive optimal vessel analysis (NOVA)-qMRA uses TOF and phase contrast (PC) sequences. The synergy between TOF and PC sequences not only provides detailed anatomical localization but also highly accurate flow quantification of the major cerebral vessels (in mL/min), making it a superior method for assessing intracranial arterial VFRs (Figure 1). This way, NOVA enables us to study cerebral collateral pathways and quantify their activation [11,12,13]. The duration of the MRI procedure typically takes about 30 to 60 min depending on the number of intracranial vessels being assessed. After the acquisition of the imaging sequences, NOVA-qMRA requires the use of dedicated post-processing software installed on an external workstation used by a trained physician to compute the VFRs on patient-specific relevant vessels.

Due to the complexity and interindividual variability in the angioanatomy of patients with bilateral ICA-stenosis and/or -occlusion, a tailored treatment strategy is required [14]. Besides clinical presentation, flow analysis might be useful as an additional parameter to aid the therapeutic decision-making. In this study, we report on a series of patients with bilateral steno-occlusive carotid disease where flow and collateral patterns were studied via NOVA-qMRA.

## 2. Materials and Methods

Kantonale Ethikkommission (Kanton Zürich, Switzerland) approval was obtained (KEK-Nr: PB_2023_01011) and all patients signed an informed consent form for the sharing and scientific diffusion of their clinical and radiological data.

### 2.1. Patient Cohort

Patients with bilateral ICA-stenosis ≥50% who received NOVA-qMRA between January 2019 and January 2022 were included in this study. The severity of ICA steno-occlusive disease was classified according to the North American Symptomatic Carotid Endarterectomy Trial (NASCET) criteria, a widely used classification system based on measuring the degree of arterial narrowing caused by atherosclerotic plaques, with the aim of guiding clinical decisions regarding medical or surgical treatment [15]. According to the NASCET criteria, the degree of stenosis is calculated as a percentage reduction in the lumen diameter of the ICA by comparing the residual lumen diameter at the site of maximal stenosis to the normal lumen diameter of the distal ICA, which serves as the reference point. Patients were divided into two groups, i.e., treated and non-treated, depending on whether they were revascularized or not. Demographic, clinical, radiological, and therapeutic data of the included patients during the hospital stay were collected. The National Institutes of Health Stroke Scale (NIHSS) and modified Rankin scale (mRS) at the time of hospital admission and discharge were collected to document functional status and outcome. The Trial of Org 10172 in the acute stroke treatment (TOAST) system was used to identify the likely etiology of the stroke of the included patients. This system does not specify nor quantify the severity of the intracranial atherosclerosis, but it does categorize ischemic strokes into 5 subtypes based on the underlying mechanism of injury [16].

### 2.2. Imaging Data

The volume flow rates (VFRs) of the following arterial segments were collected: A2-segment of the anterior cerebral artery (A2-ACA), M1-segment of the middle cerebral artery (M1-MCA), and P2-segment of the posterior cerebral artery (P2-PCA). The hemispheric VFR (hVFR) was calculated as the summation of the ipsilateral A2-, M1-, and P2-VFRs [16]. The choice of the arterial segments provided a comprehensive assessment of the functionality of both the anterior (via ACoA) and posterior (via PCoA) collateral pathways. These segments were specifically targeted because they serve as critical “bridges” in redistributing blood flow in the presence of steno-occlusive disease, offering insights into collateral circulation. The VFRs were compared between the treated- and non-treated subgroups. Within the treated group, the VFRs of the revascularized vs. non-revascularized hemisphere were also compared (Figure 2).

In symptomatic patients, the timing at which the NOVA-qMRA was performed was defined in the acute phase (0–7 days), subacute phase (8 days to 6 weeks), and chronic phase (>6 weeks) from symptoms’ onset [17].

### 2.3. Statistical Analysis

Statistical analysis was performed using R Studio (2023.06.2). After assessing the normal distribution of the data using the Shapiro–Wilk test, VFRs of the segments A2, M1, and P2 were analyzed and compared, either in an intra- or intergroupal manner. For intragroupal comparisons, one-tailed, paired *t*-tests were conducted, and for intergroupal comparisons, one-tailed, unpaired *t*-tests were performed. A *p*-value < 0.05 was considered to represent statistical significance.

## 3. Results

### 3.1. Demographic and Clinical Data

In total, 22 patients (1 woman and 21 men, mean age ± SD: 68 ± 10 years) with bilateral ICA-stenosis and/or -occlusion were included from our prospective register. Twenty patients presented severe atherosclerosis defined as Grade 1 according to the TOAST classification and two presented with TOAST Grade 2 [18]. Further clinical and demographic data are summarized in Table 1.

At the time of hospital admission, only one patient (4.5%) presented with an NIHSS > 7, and two patients (9%) had an mRS > 2. At discharge, two patients (9%) showed an NIHSS > 7 and one patient (4.5%) presented an mRS > 2. Twelve patients (55%) received surgical treatment, of which four (33%) underwent carotid endarterectomy (CEA), seven (58%) underwent flow augmentation superficial temporal artery to middle cerebral artery (STA-MCA) bypass, and one patient (8%) underwent a combined approach consisting of CEA of the left external carotid artery (ECA) followed by ipsilateral STA-MCA bypass. One patient (5%) received carotid artery stenting. Nine patients (41%) were managed conservatively. The therapeutic strategies are summarized in Table 2.

### 3.2. Hemodynamic Assessment

Nineteen patients (86%) presented with symptoms. The timing of the NOVA-qMRA after the cerebrovascular insult (CVI) is shown in Table 3. In three patients (14%), bilateral ICA-stenosis was discovered incidentally.

### 3.3. Treated Group

Thirteen patients received a cerebral revascularization procedure based on the NOVA-qMRA VFRs co-adjuvate by the clinical presentation. In the treated group, two patients (15%) presented with bilateral ICA-stenosis ≥ 70%, three (23%) with bilateral ICA-occlusion, five (38%) had unilateral ICA-occlusion and contralateral stenosis ≥ 70%, and three had unilateral ICA-occlusion and contralateral ICA-stenosis ≥ 50% and <70%. Out of the treated patients, 11 (84%) exhibited symptoms. Before treatment, the mean VFR of the hemisphere selected for revascularization was 253 mL/min, whereas the mean value of the contralateral hemisphere was 304 mL/min. The mean M1-VFR on the side selected for revascularization was significantly lower than that of the contralateral side (88 mL/min vs. 130 mL/min, *p*-value of 0.01). Please refer to Table 2 for an overview of the therapeutic strategy and the chosen revascularized side in this subgroup of patients, and to Table 4 for a visual summary of the segmental VFRs in the treated group.

### 3.4. Non-Treated Group

Nine patients were treated conservatively. Among these, two patients (22%) presented with unilateral ICA-occlusion and contralateral stenosis ≥ 70%, while the remaining seven patients (78%) had unilateral ICA-occlusion and contralateral ICA-stenosis ≥ 50% and <70%. In the non-treated group, eight patients (89%) were symptomatic. In this subgroup, no significant differences were observed after comparing the segmental VFRs of the right and left sides (see Table 5). The mean hemispheric VFR of the left side was 292 mL/min, whereas the mean VFR of the contralateral side was 305 mL/min.

### 3.5. Comparison Between Treated and Non-Treated Groups

When comparing the segmental VFRs between the treated and non-treated groups, the M1-VFR was found to be significantly higher (*p*-value = 0.05) in the non-treated subgroup. Furthermore, the P2- and A2-VFRs were found to be significantly higher (*p*-values of 0.04 and 0.03, respectively) in the treated subgroup.

## 4. Discussion

This study highlights the potential value of NOVA-qMRA in evaluating blood flow and collateral circulation in patients with bilateral steno-occlusive carotid disease. Managing these patients is particularly challenging, as their clinical presentation can vary widely and is often nonspecific [19,20,21]. Incorporating flow data derived from NOVA-qMRA into clinical assessments could enhance therapeutic decision-making, especially in cases where the laterality of symptoms is ambiguous. This information may prove invaluable in guiding the choice of the hemisphere to revascularize [8,17,18,19,20,21,22,23,24].

We analyzed NOVA measurements from 22 patients with bilateral internal carotid artery (ICA) stenosis and/or occlusion, 19 of whom presented with neurological symptoms. Among these patients, 13 underwent revascularization. Flow velocity and volume flow rates (VFRs) were measured in key arterial segments, defined as A2, M1, and P2. In the treated group, the segmental VFR analysis revealed a significantly lower M1-VFR on the revascularized side compared to the contralateral side (88 ± 53 mL/min vs. 130 ± 56 mL/min, *p* = 0.01). Conversely, in the non-treated subgroup, there were no significant differences in M1- or P2-VFRs between the two hemispheres.

Interestingly, the P2-VFRs were significantly higher in the treated group compared to the non-treated group. This finding could reflect the increased activation of the collateral leptomeningeal pathways through the posterior cerebral artery [16,25,26]. Such collateral flow activation may serve as a compensatory mechanism in response to diminished anterior circulation flow, further underlining the utility of NOVA-qMRA in identifying critical hemodynamic patterns.

The clinical outcomes in our cohort were encouraging. At the time of hospital discharge, among the patients included in the analyzed cohort, only one presented a neurological worsening, while the rest of the cohort did not show any relevant neurological exacerbation [25,27,28]. Furthermore, our findings suggest that higher M1-VFRs and lower P2-VFRs are associated with better-preserved cerebral hemodynamics and flows [16,18,27,28,29]. These observations align with the growing understanding of the importance of collateral circulation in maintaining adequate cerebral perfusion in the context of ICA disease. However, there are some limitations to this study that should be acknowledged. First, the small sample size limits the statistical power and generalizability of our findings. Larger studies are needed to validate and extend these results. Second, the study cohort was recruited from a single tertiary referral center, which introduces an inherent selection bias and may not reflect the broader patient population. Furthermore, this study represents a retrospective analysis that aimed to provide initial data on the implementation of NOVA-qMRA in the study of anterior cerebral circulation, and it did not provide a thorough comparison of NOVA-qMRA with other imaging techniques in this cohort of patients; thus, the power of the analysis is reduced. Despite these limitations, this pilot study provides important preliminary evidence supporting the integration of NOVA-qMRA into the diagnostic workup of patients with bilateral steno-occlusive carotid disease.

## 5. Conclusions

This study endorses the added value of NOVA-qMRA to the diagnostic workup process of patients with bilateral steno-occlusive carotid disease. The analysis of segmental VFRs, especially of M1- and P2-VFR, might be helpful to select and tailor revascularization treatment.

## Figures and Tables

**Figure 1 brainsci-15-00211-f001:**
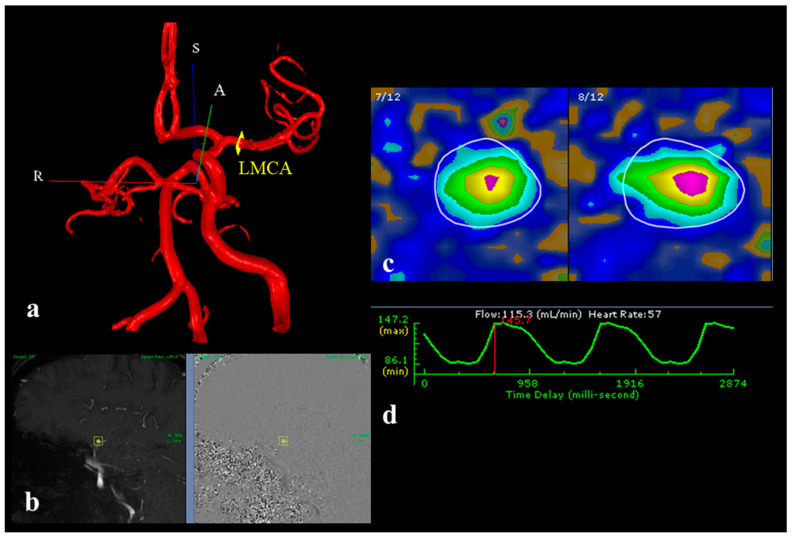
Workflow of the imaging post-processing in the NOVA software (VasSol, Chicago, IL, USA) The post-processing starts with the surface rendering of the cerebral vasculature from the 3D TOF MRA. A region of interest (ROI) is then placed perpendicular to the axis of the vessel of interest; in this exemplifying case, it is the left MCA. In this Figure R (red line) means Right, S (blue line) means Superior and A (green) means anterior: the axes are given for three-dimensional clarity (**a**). The cut is then adjusted according to the gated 2D phase contrast sequences (**b**) and proofed with the velocity contouring of the vessels, where the greatest intravascular velocity is portrayed in purple and the concentric slower velocity and progressively identified by yellow, green and light blue (**c**). The corresponding waveform, including the volume flow rate of the specific vessel, is automatically obtained (**d**).

**Figure 2 brainsci-15-00211-f002:**
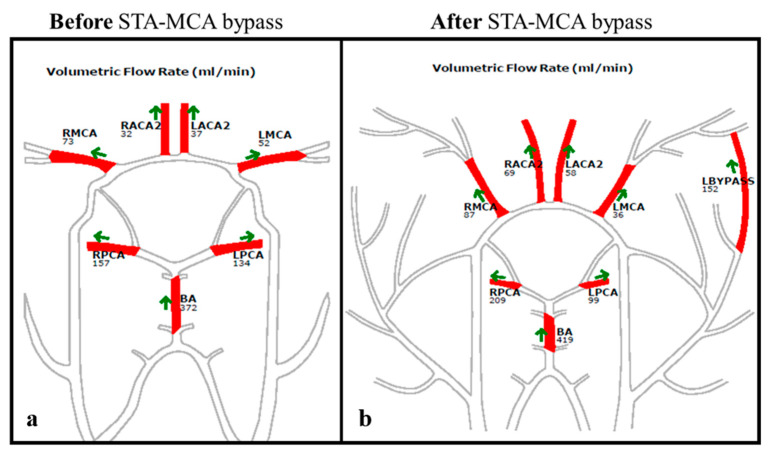
NOVA-maps of a 67-year-old patient who presented with an acute left ICA-occlusion and contralateral chronic ICA-occlusion. On the **left**, the preoperative NOVA-map shows a reduced left MCA-M1-VFR of 52 mL/min (the normal range for LMCA VFR in subjects above 61 years of age is between 81 mL/min and 201 mL/min) and an increased left PCA-P2-VFR of 134 mL/min (the normal range for LPCA VFR in subjects above 61 years of age is between 34 mL/min and 88 mL/min) (**a**); on the **right** is the NOVA-map 2 days after an acute left flow augmentation STA-MCA bypass showing adequate VFR of the bypass and a reduction in the LPCA VFR, demonstrating less need for collateral pathways (**b**).

**Table 1 brainsci-15-00211-t001:** Clinical data of included patients with bilateral ICA-stenosis and/or -occlusion (all patients n = 22).

	*n* (%)
Male gender	21 (95)
Symptomatic disease	19 (86)
Unilateral ICA-occlusion + contralateral stenosis ≥ 70%	7 (32)
Bilateral ICA-stenosis ≥ 70%	2 (9)
Bilateral ICA-occlusion	3 (14)
Unilateral occlusion and contralateral stenosis ≥ 50%, <70%	10 (45)
TOAST 1	20 (91)
TOAST 2	2 (9)
Additional Basilar artery stenosis	2 (9)
Additional unilateral vertebral artery stenosis or occlusion	4 (18)
Additional bilateral vertebral artery stenosis or occlusion	1 (5)

ICA = internal carotid artery; TOAST = trial of ORG 10172 in acute stroke treatment.

**Table 2 brainsci-15-00211-t002:** Therapeutic strategies and localization.

Therapy	*n* (%)		Bilateral	Left Side Only	Right Side Only
Conservative	9 (41)		NA	NA	NA
Endovascular	1 (5) *		0	1	0
Surgery	12 (55)		1	8	3
	4	CEA			
	7	STA-MCA bypass			
	1	Combined approach ^1^			

CEA = carotid endarterectomy, STA-MCA bypass = superficial temporal artery-middle cerebral artery bypass, ^1^ Combined approach = CEA of the left ECA (external carotid artery), and ipsilateral STA-MCA bypass *: carotid artery stenting (CAS), NA = Not applicable.

**Table 3 brainsci-15-00211-t003:** Timing of NOVA-qMRA after CVI for the 19 symptomatic patients.

	Acute (0–7 Days)	Subacute (8 Days–6 Weeks)	Chronic (>6 Weeks)
Timing of NOVA-qMRA after CVI	9 (47%)	5 (26%)	5 (26%)

NOVA-qMRA = non-invasive optimal vessel analysis quantitative magnetic resonance angiography; CVI = cerebrovascular insult.

**Table 4 brainsci-15-00211-t004:** Comparison between segmental VFRs (A2, M1, P2) on the hemisphere selected for revascularization and segmental VFRs of the contralateral side.

VFR (mL/min)	Side Chosen for Revascularization (Mean ± SD)	Contralateral Side (Mean ± SD)	*p*-Values
A2	63 ± 33	68 ± 30	0.22
M1	88 ± 53	130 ± 56	0.01
P2	108 ± 41	105 ± 66	0.45

**Table 5 brainsci-15-00211-t005:** Segmental (A2, M1, P2) VFRs of the non-treated subgroup.

VFR (mL/min)	Left Side (Mean ± SD)	Right Side (Mean ± SD)
A2	78 ± 11	82 ± 27
M1	127 ± 35	150 ± 48
P2	84 ± 44	83 ± 34

## Data Availability

Data are available upon request to the corresponding author for scientific purposes only. The data are not publicly available due to data protection and privacy.

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
