# Peer review of "The Value of Non-Invasive Optimal Vessel Analysis Quantitative Magnetic Resonance Angiography for Studying Flow and Collateral Patterns in Patients with Bilateral Carotid Steno-Occlusive Disease"

_brainsci, 2025, doi:10.3390/brainsci15020211_

Round 1
Reviewer 1 Report
Comments and Suggestions for Authors
Major issue: This article aims to present the value of a new sequence for detecting cerebral flow in patients with ICA stenosis. However, the results only showed a simple comparison of cerebral flow between the treated and non-treated subgroups, as well as before and after treatment subgroups, without demonstrating the advantages of this technique. It is recommended to compare this technique to the gold standard methods for detecting blood flow so that readers can understand the precision and advantages of this new sequence.
Minor issues:
- In the abstract, the full name of NOVA-qMRA should be mentioned the first time it is introduced.
- The sample size is small.
- In the introduction, it is advisable to highlight the advantages of NOVA-qMRA compared to other methods for detecting cerebral flow, such as its non-invasiveness.
- Please provide the full name when introducing abbreviations for the first time (e.g., IRB in line 56).
- In the methods section, a detailed description of the NASCET criteria may help clarify the classification used in this study.
- The article only focuses on specific segments of the arteries in the Willis circle (A2-ACA, P2-PCA, and M1-MCA). The rationale for this selection should be explained in the methods section.
- For figure 1, it would be helpful to indicate the segments and their VFR in the figure legend.
- Some of the data in the study did not follow a normal distribution, making it inappropriate to use T-tests for all data.
- The TOAST classification mentioned in the results section (line 97-98, table 1) should also be described in the methods section.
Author Response
Major issue: This article aims to present the value of a new sequence for detecting cerebral flow in patients with ICA stenosis. However, the results only showed a simple comparison of cerebral flow between the treated and non-treated subgroups, as well as before and after treatment subgroups, without demonstrating the advantages of this technique. It is recommended to compare this technique to the gold standard methods for detecting blood flow so that readers can understand the precision and advantages of this new sequence.
Dear colleague, the primary aim of this retrospective analysis was an investigation on the benefits of the use of NOVA-qMRA in a complex subgroup of patients with severe intracranial steno-occlusive disease. We believe that the presented data, though preliminary and retrospective in nature, does show the added value of NOVA-qMRA in the diagnostic and therapeutic workup of these complex patients by demonstrating an improvement of the VFRs in the follow-up. Indeed, this study aimed at a simple comparison: it might seem reductive, but it represents a first attempt to endorse the implementation of this advanced imaging technique to study the anterior Willis. However, we do understand your comments and agree that this analysis lacks some power and a degree of objectiveness. These considerations have been now listed among the limitations of the study.
Minor issues:
- In the abstract, the full name of NOVA-qMRA should be mentioned the first time it is introduced. -> The abbreviation is now specified in the Abstract. Thank you for highlighting it.
- The sample size is small -> Dear colleague, this is indeed a major limitation of the present study, and we have highlighted it in the Discussion.
- In the introduction, it is advisable to highlight the advantages of NOVA-qMRA compared to other methods for detecting cerebral flow, such as its non-invasiveness -> Dear reviewer, we incorporated now more details to the brief explanation of the added value of NOVA-qMRA in Introduction. Thank you for the valuable insight.
- Please provide the full name when introducing abbreviations for the first time (e.g., IRB in line 56) -> The full name for the abbreviation was added.
- In the methods section, a detailed description of the NASCET criteria may help clarify the classification used in this study -> Dear colleague, this is indeed a relevant point: we provided now a more detailed explanation of the way the stenosis is measured according to the criteria.
- The article only focuses on specific segments of the arteries in the Willis circle (A2-ACA, P2-PCA, and M1-MCA). The rationale for this selection should be explained in the methods section -> Dear reviewer, we appreciate your input, and we have added now more information for the readers to understand the rational behind the choice of the studied arterial segments.
- For figure 1, it would be helpful to indicate the segments and their VFR in the figure legend -> Dear colleague, the caption of Figure 1 has been updated according to your suggestion.
- Some of the data in the study did not follow a normal distribution, making it inappropriate to use T-tests for all data -> Dear reviewer, we checked again the data included in the analysis and assessed their normal distribution. Therefore, we did not change the statistical analysis.
- The TOAST classification mentioned in the results section (line 97-98, table 1) should also be described in the methods section -> Dear colleague, a new concise but explanatory paragraph has been added to Materials and Methods to provide more information on the TOAST classification system.
Reviewer 2 Report
Comments and Suggestions for Authors
Thank you for the opportunity to review the manuscript titled "The value of Non-invasive Optimal Vessel Analysis quantitative MRA for studying flow and collateral patterns in patients with bilateral carotid steno-occlusive disease" This study evaluates the application of NOVA-qMRA in assessing blood flow and collateral circulation in patients with bilateral carotid steno-occlusive disease, emphasizing its potential to inform revascularization decisions. The findings suggest that NOVA-qMRA can identify critical hemodynamic patterns, thereby aiding in tailored therapeutic strategies for these high-risk patients.
Major Revisions
- Provide more technical details about the NOVA-qMRA procedure to aid readers in understanding its limitations and practical applications. How long did it take? Any other imaging done for the assessment? Was the contrast used? How did MRI radiographers set up the sequence settings?
- Visual examples of the NOVA q-MRA studies are essential.
Author Response
Thank you for the opportunity to review the manuscript titled "The value of Non-invasive Optimal Vessel Analysis quantitative MRA for studying flow and collateral patterns in patients with bilateral carotid steno-occlusive disease" This study evaluates the application of NOVA-qMRA in assessing blood flow and collateral circulation in patients with bilateral carotid steno-occlusive disease, emphasizing its potential to inform revascularization decisions. The findings suggest that NOVA-qMRA can identify critical hemodynamic patterns, thereby aiding in tailored therapeutic strategies for these high-risk patients.
Major Revisions
- Provide more technical details about the NOVA-qMRA procedure to aid readers in understanding its limitations and practical applications. How long did it take? Any other imaging done for the assessment? Was the contrast used? How did MRI radiographers set up the sequence settings? -> Dear colleague, more information specific to the implementation of NOVA-qMRA have been added to Introduction.
- Visual examples of the NOVA q-MRA studies are essential -> Dear reviewer, could you comment further on this point? Would you suggest adding some more exemplifying cases in a separate section of the manuscript?
Round 2
Reviewer 2 Report
Comments and Suggestions for Authors
Thank you for addressing the question.
The scheme (Fig. 1) is interesting, but myself (and the reader) would be curious how the image produced directly by MRA imaging looks like. Please include.
Author Response
Comment: Thank you for addressing the question. The scheme (Fig. 1) is interesting, but myself (and the reader) would be curious how the image produced directly by MRA imaging looks like. Please include.
Response: Dear colleague, thank you for this observation. We have now added a new figure to explain the workflow of the imaging post-processing on the NOVA software.
